# Prevalence of Misophonia and Correlates of Its Symptoms among Inpatients with Depression

**DOI:** 10.3390/ijerph17155464

**Published:** 2020-07-29

**Authors:** Marta Siepsiak, Anna Maria Sobczak, Bartosz Bohaterewicz, Łukasz Cichocki, Wojciech Łukasz Dragan

**Affiliations:** 1Faculty of Psychology, University of Warsaw, 00-183 Warsaw, Poland; wdragan@psych.uw.edu.pl; 2Department of Cognitive Neuroscience and Neuroergonomics, Institute of Applied Psychology, Jagiellonian University, 30-348 Cracow, Poland; ansonsobczak@gmail.com (A.M.S.); bohaterewicz@gmail.com (B.B.); 3Department of Psychiatry, Andrzej Frycz Modrzewski Cracow Academy, 30-705 Cracow, Poland; lwcichocki@gmail.com

**Keywords:** misophonia, depression, anxiety, impulsivity, PTSD, MisoQuest, questionnaire for assessing misophonia, misophonia prevalence

## Abstract

Misophonia is an underexplored condition that significantly decreases the quality of life of those who suffer from it. It has neurological and physiological correlates and is associated with a variety of psychiatric symptoms; however, a growing body of data suggests that it is a discrete disorder. While comorbid diagnoses among people with misophonia have been a matter of research interest for many years there is no data on the frequency of misophonia among people with psychiatric disorders. This could be the next step to reveal additional mechanisms underlying misophonia. Until recently, the use of a variety of non-validated questionnaires and the dominance of internet-based studies have been also a major obstacles to a proper definition of misophonia. A total of 94 inpatients diagnosed with depression were assessed for misophonia with face-to-face interviews as well as with MisoQuest—a validated misophonia questionnaire. The prevalence of misophonia among these patients and the congruence of MisoQuest with face-to-face interviews were evaluated. Additionally, the patients filled in a series of questionnaires that measured a variety of psychiatric symptoms and psychological traits. Anxiety, depression, impulsivity, somatic pain, vegetative symptoms, post-traumatic stress disorder (PTSD) symptoms, gender, and age were analyzed in relation to the severity of symptoms of misophonia. Between 8.5 to 12.76% of inpatients with depression were diagnosed with misophonia (depending on measurement and inclusion criteria). MisoQuest accuracy was equal to 92.55%, sensitivity-66.67% and specificity-96.34%. Severity of misophonia symptoms was positively correlated to the greatest extent with anxiety. Moderate positive correlation was also found between severity of misophonia symptoms and depressive symptoms, intrusions, and somatic pain; a weak positive correlation was found between severity of misophonia and non-planning impulsivity, motor impulsivity, avoidance, and vegetative symptoms. There was no relationship between the severity of misophonia symptoms and attentional impulsivity or the age of participants.

## 1. Introduction

Misophonia is a kind of decreased sound tolerance [1,2] which significantly influences the quality of life of those who suffer from it. It is still underexplored and does not have either an International Classification of Diseases (ICD-11) or Diagnostic and Statistical Manual of Mental Disorders (DSM-5) classification. When people affected by misophonia are exposed to certain sound stimuli particular to each individual (e.g., sniffing, breathing, or tapping), they experience unwanted emotions (such as anger, disgust, irritation, or anxiety) which they usually consider excessive [3,4,5,6]. Moreover, they report physical sensations such as pressure in the chest, shoulders, head, or whole body as well as an increase in body temperature, pain, or breathing difficulties [3,7]. Although recent studies indicate that misophonia is a rather unique disorder, it is strongly associated with a variety of psychiatric symptoms [8,9,10]. Some results have indicated that anxiety symptoms act as a mediator of anger outbursts among individuals with misophonia [11]. These results have been replicated on a group of Chinese students [12]. Furthermore, anxiety symptoms turned out to be strongly related to the severity of misophonia in a study conducted on a population of Singaporean patients with various psychiatric diagnoses [13]. Moreover, patients with misophonia had significantly more symptoms of anxiety than the control group in the study of Schröder et al. [14]. We hypothesized that in our sample of patients with a diagnosis of depression, anxiety symptoms would be positively associated with severity of misophonia symptoms.

Rouw and Erfanian [5] found that post-traumatic stress disorder (PTSD) was one of the most common diagnoses in people with misophonia (occurring in 12% of cases). Remarkably, they noted that PTSD was the only comorbid disorder related to the severity of misophonia symptoms. Other studies have also found PTSD to be one of the most common comorbid disorders, being present in from 15.38% [9] to 30% [15] of cases. Its presence was associated with the severity of misophonia symptoms. Therefore, we assumed in this study that there would be a positive correlation between the severity of misophonia symptoms and PTSD symptoms, such as hyperarousal, avoidance, and intrusions.

Other research and observations on misophonia have led researchers to conclude that misophonia is strongly associated with obsessive-compulsive personality disorder (OCPD) and obsessive-compulsive disorder (OCD) [6,11,16]. The available data gives frequencies of the occurrence of OCD in people with misophonia at 2.4% [6], 11.53% [9], and 28% [15] while as many as 52.4% of people characterized with misophonia meet DSM diagnostic criteria for OCPD [6]. However, the occurrence of OCD characteristics such as obsession, compulsivity, and impulsivity [6] in misophonia is yet to be explored, as the data remain inconsistent. In a study by Cusack et al. [8], obsessive symptoms were more related to misophonia than compulsive symptoms. In contrast, the results of McKay et al. 2018 showed that, while some symptoms of OCD, e.g., ordering and harm avoidance, were higher among people with misophonia, other symptoms such as obsessions, neutralizing, and washing were lower [10]. Likewise, Schröderet al. 2013 state that impulsivity is an actual link between obsessive compulsive spectrum disorder and misophonia [6]. Taking into consideration the aforementioned reports, this study focused on impulsivity as a trait.

There is a large body of data indicating that people with OCD have increased impulsivity [17,18]; however, this relation appears to be quite complex [19]. While some aspects of impulsivity are positively correlated with OCD, others are associated negatively or not associated at all [20,21,22]. Moreover, self-report studies on people with OCD can be biased; therefore, they may not exactly reflect neurocognitive impulsivity [23]. Due to these disputes and discrepancies in the data on the role of impulsivity in OCD and because our goal was to explore the psychological construct of impulsivity rather than its comorbidity with other disorders, we excluded people with a diagnosis of OCD from this study. We suspected that, in spite of the exclusion of people with OCD, there would be a positive correlation between impulsivity and severity of misophonia symptoms.

Another disorder frequently reported among people with misophonia is depression, which was diagnosed in 22% of a sample of 50 people suffering from misophonia [15] and in 9.61% of a sample of 52 [9]. Both studies revealed a positive correlation between depression and the severity of misophonia symptoms. An internet-based study conducted on students of psychology also confirmed relationship between misophonia and symptoms of depression [11,12]. Another self-report study found that depression was reported by 13% of participants with misophonia [5]. In this study, we wanted to investigate whether in a clinical group of people with diagnoses of depression there is a positive correlation between severity of depressive symptoms and misophonia symptoms.

A growing body of data shows that being annoyed by noise is related to symptoms of anxiety and depression [24,25], while symptoms of depression has been found in a third of patients with tinnitus or hyperacusis [26]. Additionally, sound sensitivity has been shown to be associated with multiple vegetative symptoms of depression and pain [27,28,29]. In this study, we wanted to explore whether vegetative symptoms and pain are also associated with severity of misophonia symptoms.

In 2018, Queck et al. investigated factors related to the severity of misophonia symptoms in psychiatric patients, half of whom were diagnosed with depression [13]. Likewise, we decided to narrow down the research group to inpatients diagnosed with depression in order to avoid potential confounding effects due to other disorders. While there is no evidence that misophonia coexists with any particular disorder, at least 50% of people with misophonia suffer from some other psychiatric condition. Finding a trait or set of characteristics related to the severity of misophonia could shed light on its underlying mechanisms. Therefore, we explored how misophonia symptoms are related to psychological constructs rather than nosological entities in a homogenous group of psychiatric patients. We focused on attentional, motor, and non-planning impulsivity, symptoms of anxiety, alongside intrusions, hyperarousal and avoidance as components of PTSD. We also investigated the intensity of depressive symptoms, as the research group consisted of people who had been diagnosed with depression. Additionally, we controlled for somatic pain and vegetative symptoms that could impact auditory sensitivity.

Apart from the above, the main goal of this study was to assess the prevalence of misophonia among the patients with the diagnosis of depression. While there is already some data on the percentage of people with misophonia who suffer from depression, there is a gap in knowledge on how many people with depression suffer from misophonia. Revealing this relationship could be an additional cue on possible mechanisms underlying misophonia. Additionally, we aimed to assess the external validity of MisoQuest [30], a new questionnaire for assessing misophonia.

## 2. Materials and Methods

This study was approved by The Ethics Committee at the Faculty of Psychology at the University of Warsaw (Psychologiczne korelaty mizofonii u osób z depresją, 21/05/2019).

### 2.1. Participants

A total of 100 patients with a current diagnosis of depression were enrolled in the study. Data was missing for six participants, so 94 participants were analyzed. The psychiatric diagnoses were made by a team of hospital psychiatrists based on ICD-10 classification, prior to and independent of the study. Exclusion criteria were as follows: (a) cognitive impairment, (b) psychosis, (c) obsessive-compulsive disorder, (d) drug and/or alcohol addiction, and (e) experiencing a bipolar depressive episode. The reason the participants were in a patient unit was depression, although we excluded from the study participants with psychotic depression, depression in a progress of bipolar disorder, and when the state of depression was mainly a consequence of an alcoholic addiction. Patients were mainly treated for depression, but some of them had also diagnoses of personality or anxiety disorders. The time of their hospitalization was between two weeks and two months, and we recruited them with the help of their psychiatric doctors, who recommended patients who fulfilled the criteria of participation for this study.

Both male and female participants were recruited from psychiatric care wards in Kraków (Dr Jozef Babinski Clinical Hospital) and Warsaw (Institute of Psychiatry and Neurology). Participants’ ages ranged from 18 to 79 years old (*M* = 39.95, *SD* = 14.9). There was no difference in the average age (*t*(55) = 0.08; *p* = 0.935) of the men (*M* = 39.58, *SD* = 13.38) and women (*M* = 39.90, *SD* = 16.17). There were slightly more women than men in this study (45.7%; missing data for 19.1% of participants).

### 2.2. Assessments

The participants were individually evaluated by researchers trained in the diagnosis of misophonia using the measures described below.

Misophonia was identified using both the MisoQuest questionnaire [30] and a semi-structured interview based on Schröder et al.’s diagnostic criteria for misophonia [6]. During the interview, the diagnostician asked whether the patients agree or not with statements (see Appendix A) derived directly from the diagnostic criteria. After each statement, according to the situation, the researcher asked additional questions in order to eliminate possible misunderstandings and to verify whether the given symptoms have clinical meaning. MisoQuest, a 14-item self-report questionnaire for assessing misophonia, was developed and validated on a Polish population. It has excellent psychometric properties and, therefore, was suitable for the purposes of this study. Previously [30], the cutoff for clinically significant misophonia symptoms was not specified. In this study, we proposed a cutoff of 61 out of 70 points. This derives from the difference between the mean score (*M* = 65.72) and the standard deviation (*SD* = 4.3) for people classified as having misophonia in the first MisoQuest validation study conducted in 2018. The same cutoff has been chosen to identify people with misophonia in an ongoing study further assessing the external validity of MisoQuest as well as in all other studies using MisoQuest associated with the “Psychological and psychophysiological correlates of misophonia” project being conducted at the Faculty of Psychology at the University of Warsaw.

Impulsivity was evaluated with the Barratt Impulsivity Scale (BIS-11)—a 30-item self-report questionnaire which assesses three aspects of impulsivity: attentional, motor, and non-planning. The Polish adaptation was used in this study [31].

Severity of depressive and anxiety symptoms was assessed with The Hospital Anxiety Depression Scale (HADS) [32,33] and The Symptom Checklist-27-plus (SCL-27-plus). The HADS consists of 14 items, 7 of which relate to anxiety and 7 of which relate to depression. The SCL-27-plus [34,35] measures the intensity of depressive symptoms, both those currently experienced and those which have occurred throughout one’s life as well as vegetative symptoms, agoraphobia, social phobia, and somatic pain.

PTSD components were identified using the Impact Event Scale-Revised (IES-R)—a 22-item self-reported questionnaire proposed by Weiss & Marmar [36] and adapted to Polish by Juczyński and Ogińska-Bulik [37]. The IES-R is used as a screening tool for an initial diagnosis of PTSD as it contains three subscales that include the symptoms of PTSD: intrusion, hyperarousal, and avoidance. Intrusion refers to recurring images, dreams, perceptual thoughts, or impressions connected with trauma. Hyperarousal is characterized by heightened alertness, anxiety, impatience, and difficulties concentrating. Avoidance manifests in attempts to eschew thoughts, emotions, or conversations connected to the trauma.

### 2.3. Data Analysis

Patients with misophonia were identified based on MisoQuest results and face-to-face interviews in order to evaluate the prevalence of misophonia as well as the external validity of MisoQuest. Next, the sensitivity, specificity, and accuracy of MisoQuest were calculated with exact Clopper–Pearson confidence intervals.

In order to check the difference in severity of misophonia symptoms between men and women, a *t*-test was conducted on the entire sample (94 participants). Next, Pearson correlations between the severity of misophonia symptoms (measured by MisoQuest) and all the factors described above, as well as the age of the participants, were calculated on the entire sample.

Row data is provided in Appendix A.

## 3. Results

### 3.1. Prevalence of Misophonia among Inpatients with Depression

Eight out of 94 subjects were diagnosed with misophonia (8.5%), meeting criteria for misophonia in both face-to-face interview and MisoQuest.

According to the assumed cutoff in MisoQuest (61 out of 70 points), 11 patients should be classified as having misophonia (11.7%). However, two of them (61 and 62 points) did not meet the criteria in the face-to-face interview, and in one case, the symptoms were explained by PTSD (62 points). There were also 4 patients (52, 59, 59, and 60 points) who met criteria for mild misophonia [6] in face-to-face interviews but did not report a significant severity of misophonia symptoms on MisoQuest.

### 3.2. External Validity of MisoQuest

The data from the questionnaire—MisoQuest were compared with face-to-face misophonia diagnosis. For sensitivity, specificity and accuracy of MisoQuest, see Table 1.

### 3.3. Gender Differences in Misophonia Symptoms Severity and Correlates of Severity of Misophonia Symptoms

Women (*M* = 44.72; SD = 14.66) manifested a significantly higher severity of misophonia symptoms (*t*(74) = 2.5; *p* = 0.015; Cohen’s d = 0.57) than men (*M* = 35.52; SD = 17.48).

There was no correlation between age and severity of symptoms of misophonia. The severity of misophonia symptoms was moderately correlated with symptoms of anxiety and severity of depression. A moderate correlation was also found with intrusions and arousal, while avoidance correlated weakly. There was a weak correlation with non-planning impulsivity and motor impulsivity and no correlation with attentional impulsivity. Moderate and weak correlation with somatic pain and vegetative symptoms was found (Table 2).

## 4. Discussion

This study showed that the prevalence of people with misophonia among patients with depression ranges from 8.5 to 12.76%. Additionally, comparing MisoQuest to face-to-face interviews allowed us to verify its external validity in a population of patients, proving its high accuracy. Severity of misophonia symptoms was positively correlated to the greatest extent with anxiety. Moreover, severity of misophonia symptoms was weakly to moderately associated with a variety of symptoms of pathologies and traits (impulsivity).

A total of 8.5% of patients met the criteria for misophonia. If we included the 4 cases with mild misophonia, with very limited impact on one’s life, this would be 12.76%. This is the first study investigating the prevalence of misophonia among inpatients with psychiatric diagnoses and also the first study, on any population, to evaluate the prevalence of misophonia using face-to-face interviews. These percentages of individuals found to have misophonia are half those found in other studies [11,12]. However, it is not possible to draw a conclusion from this comparison. The difference in the percentage of people with misophonia is probably due to differences in the measurement tools used. We believe that previously used misophonia questionnaires might have also captured more general sound sensitivities and considered lower severities of symptoms to constitute misophonia.

Analysis of Pearson’s correlation revealed a moderate association between depressive symptoms and severity of misophonia. The relation between misophonia and depression and its symptoms is already supported by data from several studies [5,11,15]. Chronic stress might link these phenomena, as it is strongly associated with misophonia and, at the same time, is one of the leading causes of depression [38]. Reactions characteristic of misophonia—tension related to trigger sounds and, consequently, avoidance of many social situations—can lead to overactivity of the hypothalamic–pituitary–adrenal tract [39,40]. Although it is still hypothetical that depression and depressive symptoms can be caused by misophonia, this notion could be supported by comparing our results with those of other studies: depression appears to be more prevalent among people with misophonia than misophonia is among people with depression. Nonetheless, it is highly possible that this difference is due to the use of different diagnostic criteria (MisoQuest vs. the Misophonia Questionnaire) as well as the way the data was gathered (internet-based study vs. face-to-face interviews). Therefore, the exploration of the incidence of misophonia, measured with comparable criteria and questionnaires, among homogenous groups of patients as well as in the general population should be done in the future. This could help to reveal the direction of the relation between misophonia and depression as well as other disorders.

This is yet another study indicating the meaningful role of anxiety in misophonia. Anxiety was correlated to a greater degree with misophonia severity than were any of the other measured variables. These results are similar to those of Queck et al. [13], who found that anxiety was the only significant factor in multivariate regression, predicting misophonia severity in a group of psychiatric patients. While the relation between misophonia and symptoms of anxiety as a pathology is already well documented, anxiety disorders (such as generalized anxiety disorder, panic disorder, or social phobia) are not the disorders most frequently found alongside misophonia and are not reported as those most significantly related with increased misophonia symptoms. The exceptions to this are OCD, in which the role in misophonia is still disputed, and PTSD, which is an event-related disorder, in contrast to other anxiety disorders where the more diffuse nature of the anxiety facilitates interaction with misophonia. At the same time, there is also significant data on the role of increased neuroticism in misophonia [14,41]. As the role of anxiety in predicting the severity of misophonia symptoms seems to be already well described, a deeper investigation into anxiety as a trait and personality characteristics such as neuroticism versus anxiety disorders could be the next step in unveiling the mechanisms of misophonia related to anxiety.

Excluding OCD patients allowed us to assess self-described impulsivity without the risk of overlap. As a result, unlike attentional impulsivity, non-planning impulsivity and motor impulsivity were weakly but significantly correlated with misophonia symptoms. Notably, attentional impulsivity was one of only two variables (the other being age) in this study that were not correlated with the severity of misophonia; the other aspects of impulsivity were much less associated with severity of misophonia than were all the other variables. Hence, the role of impulsivity was undermined in this study, but nonetheless, strong conclusions about impulsivity should not be drawn—this relationship needs further exploration.

Regarding symptoms of PTSD, intrusions and arousal were moderately associated with misophonia symptoms while avoidance was only weakly correlated. There were 6 participants (7.4%) who reported that they had been diagnosed by their psychiatrist with PTSD; their average score on MisoQuest was 10 points higher (*M* = 48.83, *SD* = 12.37) than patients without such a diagnosis (*M* = 38.52, *SD* = 16.85). The increased severity of misophonia symptoms in people with PTSD has already been reported by Rouw and Erfanian [5]. Unfortunately, our study did not include enough participants diagnosed with PTSD to conduct a statistical analysis and to draw reliable conclusions. Nonetheless, no patients with PTSD were diagnosed with misophonia in the face-to-face interviews. Therefore, our results undermine the potential relationship between misophonia and PTSD. These results might rather indicate relations between more general sound sensitivity and psychological distress and hyperarousal [42,43].

Vegetative symptoms were found to be weakly yet significantly correlated with the severity of misophonia symptoms, while somatic pain was found to be moderately significantly correlated. Although many studies discuss the vegetative symptoms experienced by people with misophonia in the presence of their misophonic triggers, to the best of our knowledge, this is the first study to measure the association between the above symptoms and misophonia severity in general. We are not surprised with these results. We expected that vegetative symptoms and somatic pain would be related to misophonia symptom severity to a certain but limited extent based on the associations between sound sensitivity and somatic disturbances and pain discussed in the Introduction section. Future research should check whether these variables are more related to general sound sensitivity than to misophonia. It is possible that, because misophonia is a more specific condition than general sound sensitivity/hyperacusis, the presence of pain and vegetative symptoms could be lower in misophonia.

Women manifested a significantly higher severity of misophonia symptoms than did men. Our results are in line with recent research conducted by Queck et al. [13] or Erfanian et al. [9]. However, we draw different conclusions to previous studies [11,12] which found no gender differences in severity of misophonia. Gender differences in accepted standards for certain scales that are used in clinical practice are usually not taken into account in research studies. As with other constructs, the standards for genders might differ also for misophonia.

The severity of misophonia symptoms was not associated with the age of participants. However, much age data was missing, which could have affected the results. In a study by Queck et al. [13] on patients with psychiatric diagnoses, a negative correlation between the severity of misophonia symptoms and age was observed, but only before adjusting for confounding factors. Similarly, there was no relation with age in the study by Wu et al. [11]. Nonetheless, the data are inconclusive: in the study conducted by Rouw & Erfanian [5], while severity of misophonia symptoms decreased with age and there was negative correlation with a small effect between age and severity of misophonia, they noted that it increased in the two oldest age categories (over 65). Thus, the association between age and severity of misophonia needs further exploration.

Assessment of MisoQuest’s additional external validity showed that, with its high accuracy, it is a good tool for assessing misophonia. It has lower sensitivity than specificity, which means that there is a risk of false negative cases. In fact, 4 patients diagnosed in face-to-face interviews with misophonia with limited impact on one’s life did not reach the cutoff for misophonia in MisoQuest. Results slightly below or above the cutoff for misophonia should be evaluated carefully, especially in populations of patients with psychiatric diagnoses. MisoQuest was created with an aim to identify people whose quality of life is significantly lower because of misophonia [30], and apparently, it captures more severe cases. Further assessment of validity of MisoQuest is needed on the general population.

### Limitations

This study has two meaningful weaknesses. The major drawback of the study is the significant amount of missing data (age and gender) that was a result of mistakes during the data collection. Another limitation is that caution must be used when comparing these data with those of other studies (especially studies in which only questionnaires were used, with no face-to-face interviews). This is because MisoQuest is based on a slightly different approach to misophonia than the Misophonia Questionnaire (MQ), which has, to date, been used in the majority of studies. At the time of conducting this study, there was no Polish version of the MQ available. In the future, it would be worthwhile to compare characteristics of people diagnosed with misophonia by those two different questionnaires.

## 5. Conclusions

The prevalence of misophonia among patients with depression was evaluated in this study and stands at 8.5–12.76%. Additionally, the results show that MisoQuest can be helpful for identifying misophonia among patients with psychiatric diagnoses. The severity of misophonia symptoms was associated weakly to moderately with various symptoms and psychological characteristics and was the most associated with anxiety. Anxiety, however, is a more general psychological construct that not only is related to multiple psychiatric conditions but also for which intensity differs based on the temperamental characteristics of individuals. The aforementioned results are in line with other studies, suggesting that misophonia might be an independent disorder.

## Figures and Tables

**Table 1 ijerph-17-05464-t001:** Sensitivity, specificity, and accuracy of MisoQuest.

MisoQuest	Value %	(95% CI)
Sensitivity	66.67	(34.89–90.08)
Specificity	96.34	(89.68–99.24)
Accuracy	92.55	(85.26–96.95)

**Table 2 ijerph-17-05464-t002:** Correlations with severity of misophonia symptoms assessed with MisoQuest.

	Age	HADS Anxiety	BIS Attentional	BIS Motor	BIS Non-Planning	SCL Vegetative	SCL Pain	HADS Depression	IES Intrusions	IES Arousal	IES Avoidance
MisoQuest Pearson’s correlation *p*-value	0.024	0.444	−0.178	0.205	0.204	0.276	0.307	0.325	0.324	0.325	0.263
	0.856	0.000	0.086	0.047	0.049	0.007	0.003	0.001	0.001	0.001	0.010

HADS—Hospital Anxiety and Depression scale; BIS—Barrat Impulsivity Scale; SCL—Symptom Checklist Scale; IES—Impact of Event Scale.

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
