# Peer review of "Prevalence of Misophonia and Correlates of Its Symptoms among Inpatients with Depression"

_ijerph, 2020, doi:10.3390/ijerph17155464_

Round 1

Reviewer 1 Report

This manuscript examines misophonia, anxiety, and impulsivity among psychiatric inpatients diagnosed with depression. Misophonia is understudied, and this is the first study to diagnose misophonia with a face-to-face interview and to collect data on misophonia prevalence among diagnosed psychiatric patients. Another strength is that it employed a validated misophonia scale of the authors’ own creation.  Thus it makes several novel contributions to the literature; I have a few suggestions for improving the manuscript.

As noted, this is the first study of misophonia among psychiatric patients; this is an unusual and valuable sample and more information is needed about them. First, though they were included in the study for a current diagnosis of depression, it is not clear whether depression is why they were admitted to an inpatient unit. What diagnoses were the patients admitted to the wards for? What other diagnoses did they have?  How long were they in the hospital?  What was the procedure for recruiting patients into the study?  A table of demographic information and psychiatric characteristics of the sample would be helpful.

Another unique attribute of the study is that misophonia was diagnosed with a face-to-face interview. More information is needed about the interview used for diagnosing misophonia, as well as the diagnostic criteria used. Was it a structured or unstructured interview? Is there any data on the reliability of the interview? What are the criteria for “mild misophonia"?

The authors need to keep the unique characteristics of the sample in mind when interpreting their data. For example, the lack of gender differences in anxiety in Table 1 is very hard to interpret, since in most samples one would expect to find a gender difference in trait anxiety.  Is the lack of difference a chance finding? Is lack of gender differences in anxiety common among a Polish demographic? Is this finding because depression, or their inpatient status, results in uniformly high anxiety, thus reducing variance?  Because the gender differences in particular are not easily interpretable, my advice is to take the gender difference section out of the paper and focus on the interrelationships between misophonia and the other measures.

The regression analysis is also difficult to interpret, in part because not enough information was presented about the analytic procedures. What kind of multiple regression was performed? Were all 3 variables simultaneously entered, or was a stepwise procedure used?  The lack of an effect for depression could be due to reduced variance for depression (since it was an inclusion criterion for the study), or because of a correlation between depression and anxiety.

I do not understand the following sentence near the beginning of the Discussion: “Furthermore, participants diagnosed with depression did not manifest misophonia more frequently than a population of students”.  Does this mean that the participants with depression did not manifest misophonia more frequently than the students in the cited sample (e.g., that depression may be protective for misophonia)?  Or that participants in this study diagnosed with depression did not manifest misophonia more frequently than did a population of students with depression?

The authors need to check their data and data analyses for accuracy. According to Table 1, the HADS Depression scores are listed as not having a significant gender difference (with p=1.000!); however, a quick t-test calculated based on the means, SDs, and sample size in the table finds that these scores are in fact significantly different (p<.0001).  Another potential error is that the mean and SD are the same for the women's HADS-Depression and IES-Intrusions scores. All means and statistical analyses in this table should be checked.  Further, given these problems, the authors should check the accuracy of the HADS and IES correlations in Table 2, which are very close to each other (or in one case identical).  There must be some mistake in the Table, data analysis, or dataset.  

The authors state (with regard to studies of misophonia, presumably) that “to the best of our knowledge, this is the first time anxiety has been explored as a trait, rather than as a symptom of pathology.” Yet, in the next paragraph (and through much of the Introduction), the authors discuss anxiety as a symptom of pathology, going into great detail about many different anxiety disorders.  If the manuscript’s intention is to focus on anxiety as a trait, then much of this review of anxiety disorders and misophonia, while interesting, is not relevant to the manuscript (in fact, both the Introduction and Discussion are overly long).  On the other hand, one can make a very different argument about how anxiety should be conceptualized in the current study:  anxiety is a frequent symptom in individuals with depression, and all of the participants in this study were diagnosed with depression.  Thus, it may be that, compared to other studies of misophonia in student groups without diagnostic information, the current study is more relevant to studying anxiety as a symptom of pathology.

On a related issue, the authors should be more mindful of their description of studies throughout the Introduction and Discussion as involving clinical diagnoses versus symptoms of psychopathology in unselected samples.  Here are two example sentences from the Discussion: “in this study on people with diagnosed depression, we found that the rate of occurrence of misophonia was half that found among students of psychology” and “The increased severity of misophonia symptoms in people with PTSD has already been reported by Rouw and Erfanian (2018).” However, none of the studies cited in these sentences involved diagnoses of depression or PTSD.  Indeed, one of the primary strengths of this study is that it is the first to examine the prevalence of misophonia among inpatients with a confirmed psychiatric diagnosis.  

Another strength of this study is that it is the first study to evaluate the prevalence of misophonia using face-to-face interviews. Given this important contribution, rather than just saying “MisoQuest correctly assessed the presence or absence of misophonia in 87 patients (91%)”, give both the sensitivity and specificity. The authors also might note that, since the interview is evaluating proposed criteria for misophonia, we do not know what whether the diagnostic interview or the MisoQuest is "correct". It may be the MisoQuest is more valid than the diagnostic criteria at identifying misophonia.

Reviewer 2 Report

Thankyou for asking me to review the manuscript "Prevalence and correlates of misophonia severity among inpatients with depression". 

The authors have raised an interesting however there are several points that needs to be raised with regard to this manuscript and the methodology:

  • There are no description of- or clear research questions or hypothesis. It remains unclear what the study aim is and why the chosen instruments are used.
  • The IMRAD structure of manuscripts are not always followed, eg. there are methodological considerations described in the introduction section.
  • The design and the methods for including participants is difficult to follow, eg. why was it not possible to obtain gender in all the participants? In the discussion the authors write "This is the first study investigating the prevalence of misophonia among inpatients with psychiatric diagnoses", is prevalence of misophonia an aim?
  • The content of the MisoQuest questionnaire is not described.

  • There were only eight participants that were classified with misophonia in a sample of depressed patients recruited from psychiatric care wards, and I am not sure whether this is sufficient numbers to perform valid statistical analyses? Have you consulted a statistician?
  • Table 1 of the results are interesting, however since the aims are not described, it remains unclear why gender differences are reported on the total sample?
  • The paper overall is well written, however the design and the methods it now is presented does unfortunately not seem appropriate to support the findings.
  • Last, it is not clear how- or in what way the results can be implemented in a clinical or research setting nor be applied to other populations?
  • It is absolutely interesting, and the authors hava done a solid amount of work, but I think that the manuscript has to be rewritten and the above points clarified before it can be considered for publication.

Round 2

Reviewer 2 Report

The manuscript has been improved and I have no more comments.